# An Incomplete Loop: Instruction Inference, Instruction Following, and In-context Learning in Language Models

**Emmy Liu & Graham Neubig**
Language Technologies Institute
Carnegie Mellon University
emmy@cmu.edu, gneubig@cs.cmu.edu

**Jacob Andreas**
CSAIL
Massachusetts Institute of Technology
jda@mit.edu

## Abstract

Modern language models (LMs) can learn to perform new tasks in different ways: in *instruction following*, the target task is described explicitly in natural language; in *few-shot prompting*, the task is specified implicitly with a small number of examples; in *instruction inference*, LMs are presented with in-context examples and are then prompted to generate a natural language task description before making predictions. Each of these procedures may be thought of as invoking a different form of reasoning: instruction following involves deductive reasoning, few-shot prompting involves inductive reasoning, and instruction inference involves abductive reasoning. How do these different capabilities relate? Across four LMs (from the gpt and llama families) and two learning problems (involving arithmetic functions and machine translation) we find a strong dissociation between the different types of reasoning: LMs can sometimes learn effectively from few-shot prompts even when they are unable to explain their own prediction rules; conversely, they sometimes infer useful task descriptions while completely failing to learn from human-generated descriptions of the same task. Our results highlight the non-systematic nature of reasoning even in some of today's largest LMs, and underscore the fact that very different learning mechanisms may be invoked by seemingly similar prompting procedures.[1].

## 1 Introduction

Suppose a friend is teaching you to cook. You watch them place a pan on the stove and heat olive oil at low heat, adding minced garlic and chili flakes to the olive oil once it gets hot. Later, you decide to make the recipe yourself, but you are out of olive oil. You hypothesize that the olive oil served to cook the garlic without burning it, and so substitute butter for olive oil, as it should fulfill the same function. Here, you learned a generalizable cooking procedure by reasoning **abductively**—finding the rule that best explains your experience (Frankfurt, 1958; Peirce, 1965). This is only one way of learning: the friend could have instead provided a recipe, from which you could have reasoned **deductively** about how to apply it in your kitchen. If you had remembered other times when you put but-

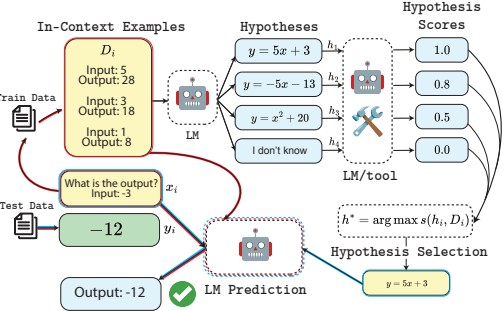

Figure 1: Diagram of abductive reasoning for an LM. Red arrows show data flow in inductive reasoning (few-shot prompting), while blue arrows show data flow in deductive reasoning (instruction following). Black arrows indicate data flow unique to abductive reasoning (instruction induction). Instruction inference generally improves on few-shot prompting and zero-shot chain of thought. However, success at inductive reasoning and success at instruction inference are not related.

---

[1]Code and data, including generated hypotheses may be found at https://github.com/nightingal3/rule_induction

| Reasoning type | Analogue in LMs | Citations |
|---|---|---|
| Deductive | Instruction following | Wei et al. (2022a); Sanh et al. (2022) |
| Inductive | In-context learning | Brown et al. (2020) |
| Abductive | Chain-of-thought (one hypothesis) | Wei et al. (2022b); Kojima et al. (2022) |
| Abductive | Externally aided instruction inference (many hypotheses) | Qiu et al. (2024); Wang et al. (2024); Zhu et al. (2023); Yang et al. (2024), |
| Abductive | LM-guided instruction inference (many hypotheses) | This paper |

Table 1: Summary of reasoning types and analogues in language models. Citations to instruction following, in-context learning, and chain-of-thought are limited to the original paper due to the high number of papers on these topics, while we have tried to list all currently available papers on instruction inference (for abductive reasoning) with LMs.

ter on a hot pan before cooking, without explicitly reasoning about why, you could have cooked the same meal **inductively**.

Each form of reasoning has a close analogue in current procedures for steering language models (LMs). In order to induce LMs to perform new tasks, we may condition them on explicit commands (**instruction following**; Wei et al., 2022a; Sanh et al., 2022), or a collection of examples from the task of interest (**few-shot prompting**; Brown et al., 2020); recently, several methods have been proposed that prompt LMs to *generate* textual commands from examples before conditioning on commands during prediction (**instruction inference**; Andreas et al., 2018; Honovich et al., 2023). But it is often unclear when to prefer one of these procedures over the other, and more generally how these different capabilities relate in current LMs. Does the ability to learn effectively from few-shot prompts imply the ability to perform instruction inference, or vice-versa? Are there tasks that can be learned from few-shot prompts, but not instructions? In this paper, we examine these questions through two tasks: learning numerical functions (§4.1) and translation models (§4.2, §4.3). We ask two questions:

**RQ1** When does (abductive) instruction inference improve LM performance over ordinary (inductive) few-shot prompting?

**Finding**: Instruction inference improves over few-shot prompting in simple cases (linear function learning and simple artificial language learning), but suffers from incorrect hypotheses in a more complex case (low-resource machine translation).

**RQ2** How does the ability to learn from instructions (deductively) relate to the ability to learn from in-context examples (inductively or abductively)?

**Finding**: The ability to learn through abduction (proposing hypotheses) is generally not related to learning through induction (few-shot learning). Deductive reasoning is generally strong, with large performance gains over using inductive reasoning alone when the provided hypothesis is correct.

Our results highlight the diverse capabilities (and diversity of different reasoning mechanisms) triggered by different prompts and examples in current LMs; future work may investigate how these reasoning types can be combined or made consistent to enhance problem solving in LMs.

## 2 Three Types of Reasoning in Language Models

We first give a formal definition of instruction following, in-context learning, and instruction inference, relating these processes to deductive reasoning, inductive reasoning, and abductive reasoning respectively. (Other ways of interacting with LMs may also evoke one or more of these forms of reasoning, but we focus on important and widely used prompting strategies.) Figure 1 gives a schematic of the reasoning loop, while Table 1 gives examples of

work falling in each category.[2] Throughout this section, we assume that we have an input $x$, and we want to produce an output $y$ with an autoregressive LM conditioned on some additional piece of data $D$ that specifies the target task: $p_{LM}(y \mid x, D)$. We use in-context learning of linear functions as a running example.

## 2.1 Instruction Following

In instruction following, we want to map each input $x$ to a $y$ according to a general instruction or prediction rule $D$ that is specified in the input. This may be viewed as a kind of deductive reasoning, which in begins with one or more premises, and applies logically valid rules to reach a conclusion (Shapiro & Kouri Kissel, 2024). For instance, the instruction $D$ might name a general function, such as *Apply this function to the input: y = 5x + 3*, followed by a specific query: *Input: -3*. This is illustrated in Figure 1.

## 2.2 Few-Shot Prompting

Few-shot prompting, by contrast, specifies the target task implicitly through examples. For each input $x$, the task specification $D$ consists of a set of $k$ training examples $D = \{(x_1, y_1), ..., (x_k, y_k)\}$. We then sample from $p_{LM}(y \mid x, D)$. Few-shot learning requires LMs to perform inductive reasoning (Hawthorne, 2021). Unlike deductive reasoning, there is no explicit premise stated, but the model must complete the task in a similar way to the examples. In Figure 1, this is pairs of inputs and outputs, i.e. *Input: 5, Output: 28*. These examples help specify the task as a numeric prediction task, as well as the identity of the specific target function.

## 2.3 Instruction Inference

Instruction inference connects instruction following and in-context learning: given examples $D = \{(x_i, y_i)\}_{i=1}^{k}$ and input $x$, we can instruct the model to generate a hypothesis $h$ about the identity of the task, i.e. an instruction describing the task associated with $D$. We may sample one hypothesis and immediately condition on it (a form of chain-of-thought prompting), or sample several and select the most promising one.

Compared to chain-of-thought, fewer studies have explored multi-hypothesis instruction induction; those that do typically rely on an external validation model rather than using the LM itself to evaluate. Our approach (in §3) has the following high-level form: after sampling $n$ hypotheses $h_1, ..., h_n$ from $p_{LM}(h \mid t, x)$, we evaluate each hypothesis by assigning a score $\text{score}(h_i, t)$ based on the hypothesis and in-context examples. We choose the best one $h^* = \arg\max \text{score}(h_i, D)$, and feed it back into the context as an instruction.[3]

Abductive reasoning is often called "inference to the best explanation" (Lipton, 2001; Douven, 2021). Suppose that the model generates the hypotheses shown in Figure 1. We may parse hypotheses and apply them back to the in-context examples, then ranking them by prediction error. Then the model simply has to follow the instruction as in Section 2.1.

## 3 Methods

We describe the implementation of instruction inference with multiple hypotheses in this section. As the other settings (few-shot, zero-shot chain-of-thought, etc) are already well understood, we do not explain them further, but include the exact prompts used to implement methods in Appendix B, F, and J.

---

[2]We note that there is some disagreement in the philosophy literature about the exact distinction about inductive and abductive reasoning. Here we use commonly-cited definitions, which happen to distinguish in-context learning from the more explicit process of verbalizing and testing hypotheses.

[3]Several recent papers have recently proposed similar processes, but called this inductive reasoning. We believe that abductive reasoning may be a more apt term in any process that includes hypothesis evaluation and selection. See Section 6 for more discussion.

| Domain | In-context examples ($D_i$) | Query example ($x_i$) | Expected answer ($y_i$) | Hypothesis example ($h_i$) |
|---|---|---|---|---|
| Functions | (-10, -213), (9, 167), (4, 67), ... | 15 | 287 | $f(x) = 20x - 13$ |
| Colours | (lug dax, blue green), (lug zup, blue yellow), (lug bluf, blue blue), ... | lug walm dax bluf | blue blue blue green green | lug → blue |
| Kalamang vocab (ek) | (That is their place., Tompat ma me muin), (I'm getting pandandus, I want to make a mat.), ... | Sakina is pouching guavas. | Sakina sarimara lawat. | guava → sarim |
| Kalamang grammar (ek) | "" | – | – | Order of subject and verb: SV |

Table 2: Examples of in-context examples, queries, and hypotheses in each domain.

**Instruction inference [Abduction with many hypotheses]** After generating $n$ hypotheses $\mathcal{H} = \{h_1, ..., h_n\}$, we explore methods for reranking them. For all experiments, $n = 5$. Generally, these reranking methods capture "fit" to training data. External validator reranking was used only in the functions domain (see Section 4.2). The other reranking methods used a language model. Given scores of each hypothesis, we choose:

$$h^* = \arg\max_{h_i \in \mathcal{H}} \mathsf{score}(h_i, D_i) \, .$$

Settings with instruction inference are referred to collectively as `instruction_inference`. We detail the score functions below:

`Verbalized confidence` We directly prompt the model to estimate the probability of the hypothesis given $D_i$. $\mathsf{score}_{\text{Verbal}}(h_i, D_i)$ is set to the model's probability estimation.

`P(data)` We use a separate LM, [4] to generate log probabilities for in-context examples given the hypothesis. $\mathsf{score}_{\text{P(data)}}(h_i, D_i)$ is the sum of log-probabilities of tokens of $D_i$.

`P(answer)` This is similar to `P(data)` and uses the same template, but $\mathsf{score}_{\text{P(answer)}}(h_i, D_i)$ only sums log-probabilities of tokens in answers $y_i$ in the in-context examples.

`External validator` For structured hypotheses, it is possible to parse them and apply them back to in-context examples, with the score as the negative error. It may not be possible to parse all hypotheses, if they are in natural language or inconsistent formats.

# 4 Domains and Evaluation

We investigate LM behavior in three domains: linear function inference, an artificial language learning task, and vocabulary + typological feature learning in the Kalamang language. We refer to the underlying task we would like to improve (output prediction, translation) as the **base task**, and the task of inferring an explicit natural language hypothesis from few-shot examples as the **abductive task**. Examples of each task are in Table 2. We also evaluate hypotheses themselves in each domain. For the exact prompts used in each domain, refer to Appendix B, F and J. Models used across all domains consisted of gpt family models (`gpt-3.5-turbo`; Brown et al., 2020, `gpt-4-turbo`, OpenAI et al., 2024) and `llama` family models (`llama-2-7b-chat`, `llama-2-70b-chat`, Touvron et al., 2023).

## 4.1 Linear Functions

Following investigations of language models' in-context learning abilities (Garg et al., 2022; Akyürek et al., 2023), we construct a dataset of 40 linear functions $f(x) = ax + b$, where $a$ and $b$ are uniformly sampled from the integers $[-20, 20]$, along with 5 test examples for each function, yielding 200 test examples. For each test example, we randomly generate 5

---

[4] `text-davinci-002` was used as the logprobs-generating LM for all models.

in-context examples of the function $(x_i, y_i)_{i=1}^5$ and one test example. The test example and inputs for in-context examples are also uniformly sampled from the integers $[-20, 20]$. We refer to this as the **functions domain**. Prompts used are given in Appendix B.

**Hypotheses** In this domain, hypotheses are proposed using all in-context examples for each test example. The model was presented with functions (and instructed to write them) in the form $y = ax + b$. For `external validator` reranking, we parse the generated hypothesis, and score it by its negative mean-squared error (MSE) when applied to the in-context examples.[5] If $h_i(x_{ik})$ represent executing the parsed hypothesis represented by $h_i$ on example $x_{ik}$:

$$s_{\text{ground\_truth}}(h_i, D_i) = \sum_{j=1}^k (h_i(x_{ij}) - f(x_{ij}))^2$$

**Evaluation** To evaluate the base task, we use 0-1 accuracy, as well as median squared errors. To evaluate hypotheses, we examine the accuracy of model-proposed $a$ and $b$ coefficients, as well as the Spearman correlation between proposed coefficients and the ground truth.

## 4.2 Simple Artificial Languages

Inspired by compositional instruction-following datasets, we generate a simple dataset where the inputs are nonce words such as *lug*, and the outputs are colour terms such as *blue* (Lake & Baroni, 2018; Lake et al., 2019). We call the expanded dataset the **colours domain**. The ruleset for this domain can be found in Appendix E. We generate 200 test examples and 800 training examples.[6] Prompts used are in Appendix F.

**Hypotheses** Following the original miniscan, we create a fixed minimal set of 5 in-context examples that contains each nonce word at least once, and from which the meaning of each nonce word can be reasonably inferred. During instruction inference, we isolate one nonce word at a time from the sentence, and have the model try to induce the vocabulary mapping for that word from 5 retrieved in-context examples containing that word. The ground truth grammar was written by the author, and consisted of production rules for each nonce term. Prompts used in the abductive and base tasks can be found in Appendix F.

**Evaluation** We also use 0-1 accuracy to evaluate the base task of nonce word translation, as well as corpus-level chrF (Popović, 2015). To evaluate the quality of the hypotheses themselves, we extract the production rules proposed by models and compare 0-1 accuracy against the ground truth. As a lenient evaluation on the "repeat" terms (see Appendix E for nonce terms and meanings), we marked a hypothesis for "repeat" terms as correct if it contained the term *repeat* or the numerals 2 and 3 respectively.

## 4.3 Kalamang Translation

For low-resource translation, we use the Machine Translation with One Book (MTOB) dataset, an English–Kalamang dataset with a grammar book (Tanzer et al., 2024). Kalamang is an extremely low-resource language with fewer than 200 speakers, and virtually no text on the web. The base task is to perform sentence-level translation in both directions, while the abductive task is to infer correct grammar features and vocabulary mappings. The dataset consisted of 100 test sentences (50 in each direction) and 400 train sentences. A ground-truth bilingual dictionary was provided (Visser, 2020). A grammar book was included as well, but to check correctness of high-level grammar inferences, we compiled a high level grammar sketch instead from WALS and GramBank features (Dryer & Haspelmath, 2013; Skirgård et al., 2023). More details about the grammar sketch can be found in Appendix K. Otherwise, we use the same experimental settings as the baseline, summarized in Appendix I.

---

[5]If a model produced an unparsable hypothesis (for instance, *I don't know*, or a generic answer like $y = ax + b$), that hypothesis was assigned a score of $-\infty$.

[6]We generate this dataset instead of using the original miniscan to mitigate memorization of the original nonce words, as well as to gain more test examples, as the original dataset has only 10.

| Domain | Deductive reasoning works on average | Abductive reasoning works on average | Hypothesis proposal works on average | Abductive reasoning related to inductive reasoning |
|--------|---------|---------|---------|---------|
| Functions | ✓ | ✓ | ✗ | ✗ |
| Colours | ✓ | ✓ | ✓ (half) | ✗ |
| Kalamang | ✓ (half) | ✗ | ✗ | ✗ |

Table 3: Summary of results. A checkmark indicates that the property held for all or almost all language models, a half-checkmark indicates a partial success for all or almost all language models, while an X-mark represents lack of success for most language models.

**Hypotheses** We split hypotheses into vocabulary and grammar. The ground truth instruction included retrieved examples from the wordlist and the grammar sketch. To induce the grammar sketch, we posed each grammar feature as a question, for instance: *What is the order of subject and verb in Kalamang?*, and sampled 5 sentence pairs at a time with the requisite parts of speech until the model proposed an answer to the question. If the model responded that it was unclear, this would repeat up to a maximum of 10 iterations. We only performed this process once, for `GPT-3.5-turbo` and `GPT-4-turbo` respectively. Each model used its self-induced grammar sketch at test time.[7] For vocabulary induction, we followed a similar process as in the colours domain.[8]

**Evaluation** We use corpus-level chrF. To evaluate the grammar sketch and vocabulary induction, we respectively compare to the ground truth wordlist and grammar sketch.

## 5 Results

In this section, we first evaluate models' concrete performance across different domains (**RQ1**). We highlight the significant improvement instruction inference offers in some cases in synthetic tasks, yet despite this, improvements are not uniform across tasks, and not attested in challenging domains like Kalamang. Our analysis also highlights an inconsistent correlation between the quality of generated hypotheses and few-shot learning success (**RQ2**), meaning that the ability to generate or follow instructions doesn't reliably predict task mastery or vice versa. These results (Table 3) suggest that while structured instructions can boost performance in simpler scenarios, their impact is less predictable in complex settings. Furthermore, the relationship between instruction induction, instruction following, and in-context learning is complex, and each capability may rely on separate unknown aspects of model architecture, training procedures, or data.

### 5.1 When Does Instruction Inference Improve Over In-Context Learning?

**How effective is using the true instruction?** In the domains we study, the ground-truth instruction tends to yield accurate results. Figure 3 displays mean accuracy in the linear functions and colours domains over six trials, sampled at different temperatures (see Appendix A) for details. Notably, in linear functions (3A), `GPT-4-turbo`'s accuracy increases to 96% from a baseline of 30%, with `GPT-3.5` also notably improving. In the colours domain, (3B), we see the true instruction also helps all models except for `Llama-2-7b`. However, this trend does not extend to the Kalamang task, where most models struggled to leverage the provided wordlist and grammar sketch effectively, indicated by chrF scores in Figure 4.

**How effective are models' induced instructions?** Self-generated instructions also improve on the baseline in many cases, with variations by domain. In linear functions, models' hypothesis induction markedly surpasses the few-shot baseline, with both the verbalized

---

[7]`Llama-2` models were found to be unable to propose grammar features with our prompts, so we used the `GPT-3.5-turbo` induced grammar sketch for these models.

[8]For Kalamang, due to computational costs, we cache the first parseable hypothesis proposed for each word and reuse it on subsequent sentences containing that word.

confidence and log-probability based reranking methods yielding comparable improvements (see Figure 3A again). For the colours domain (3B), instruction inference benefits performance, though not as strongly as in linear functions. chrF scores follow a similar trend, and are depicted in Appendix G. Interestingly, for `gpt-4-turbo` and `Llama-2-7b`, using the models' self-proposed hypotheses benefits performance more than using the ground-truth grammar for the colours language, despite the fact that self-proposed are not always correct. Unlike the two synthetic domains, inducing grammar and vocabulary items for Kalamang does not improve translation metrics in most cases.

### 5.2 How Does the Ability to Induce Instructions Relate to In-Context Learning?

**How accurate are model-generated hypotheses?** In assessing accuracy of models' self-generated hypotheses across different domains, our findings reveal significant variations in accuracy. In the linear functions domain, we plot hypotheses generated by two models in Figure 2, and list the Spearman $\rho$ (Spearman, 1904) as well as p-values of model-predicted coefficients in Table 4. `GPT-3.5-turbo` and `GPT-4-turbo`'s proposed coefficients are positively correlated with the real coefficients, and this is statistically significant. However, this level of accuracy is not observed with Llama-2 models, indicating a disparity in model capabilities. In the colours domain, most models, except `GPT-4-turbo`, tend to generate inaccurate hypotheses. The exact accuracy of proposed hypotheses for colours is shown in Appendix H. Mappings of simple vocabulary items such as *lug* tend to more accurate, with `GPT-4-turbo` achieving an 87% mean accuracy for hypotheses about this word. On the other hand, the difficulty increases with the terms which involved repeats, with `GPT-3.5-turbo` only achieving a 15% mean accuracy on *bluf*, the "repeat twice" term.

However, when extending these analyses to Kalamang, all models' performance in predicting grammar features is relatively poor, with `GPT-3.5-turbo` predicting 5/18 and 4/18 features correct respectively. Appendix L shows the correctness of each grammar feature. Vocabulary induction accuracy is also generally low, falling between 10-20% for most models in both the English to Kalamang as well as Kalamang to English directions. See Appendix M for details, as well as averaged segment-level chrF for the vocabulary hypotheses.

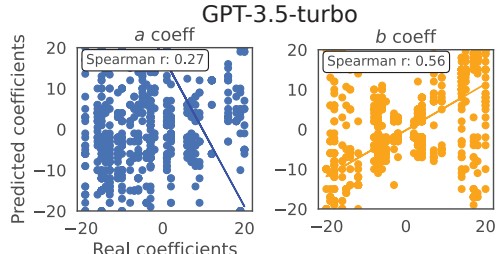

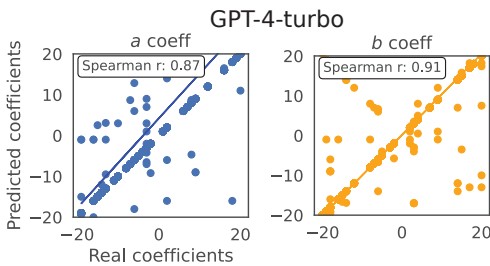

Figure 2: Real coefficients of linear functions and relationship to hypothesized coefficients for GPT-3.5-turbo and GPT-4-turbo. Remaining models can be found in Appendix D. The x-axis has been truncated for visualization purposes (as there are some large outlier hypotheses). GPT-4-turbo is able to induce a reasonable function in-context, but other models struggle.

| Model | $b$ corr. | $b$ p-val | $a$ corr | $a$ p-val |
|---|---|---|---|---|
| GPT-3.5-turbo | 0.27 | $5.0 \times 10^{-15}$ | 0.56 | $2.7 \times 10^{-67}$ |
| GPT-4-turbo | 0.85 | $2.2 \times 10^{-292}$ | 0.91 | 0.0 |
| Llama-2-7b | -0.0069 | 0.90 | 0.14 | 0.0081 |
| Llama-2-70b | 0.14 | 0.060 | -0.018 | 0.82 |

Table 4: Spearman correlation and p-values of true vs predicted coefficients for each model in the functions domain.

**Is abductive reasoning related to in-context learning ability?** In our final analysis, we examine whether the ability to induce correct instructions correlates with success in in-context learning. Specifically, we focus on the final hypotheses selected by the gpt model family, given that many hypotheses generated by `llama` models were incoherent or not

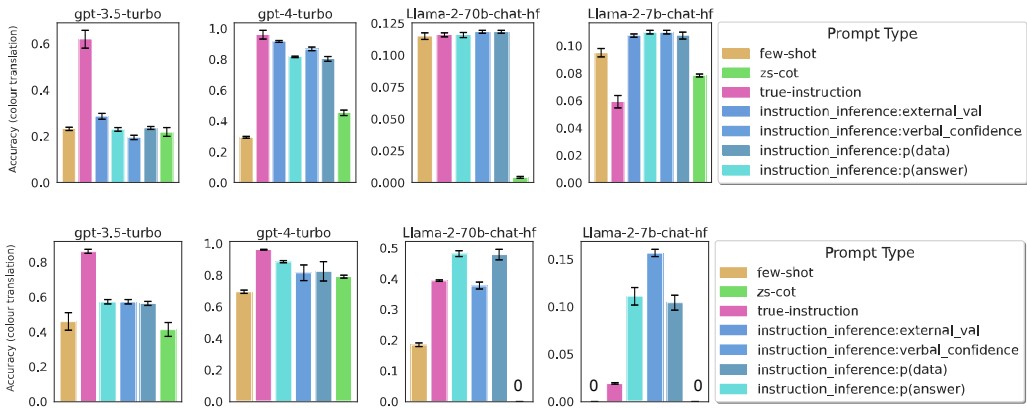

Figure 3: Accuracy of models in synthetic domains with and without hypothesis generation. Error bars indicate standard error. The top row shows results for the functions domain, while the bottom row shows results for the colours domain. Results are aggregated across 6 runs, and zero values are marked with '0'.

formatted correctly. We use the point-biserial correlation (Pearson, 1895) to assess the relationship between the accuracy of a model's final hypothesis and its success in in-context learning.

In linear functions, we examine GPT-3.5-turbo because of GPT-4-turbo's consistent accuracy in selecting correct hypotheses.[9] The analysis reveals a chance-level agreement (0.0013), suggesting GPT-3.5-turbo may be able to predict outputs without identifying the underlying function well, or vice versa. This reveals a dissociation between prediction accuracy and instruction induction.

In the colours domain, we examine both gpt models and conduct a similar analysis for each nonce word. We divide the test examples into examples containing each word, and examine the point-biserial correlation between accuracy of induced word meanings and correct translations in the few-shot context. This correlation is generally low, and few p-values are significant after correcting for multiple comparisons with false discovery rate (FDR) (Benjamini & Hochberg, 1995). See Appendix H for details.

In Kalamang, we repeat the process of computing point-biserial correlation between vocabulary induction correctness with segment-level chrF in the few-shot translations. Unlike the other domains, there is a small positive correlation between correct vocabulary hypotheses and chrF in gpt models. See Appendix M for details. We note that chrF is a more fine-grained measure than accuracy, and the initial scores were low enough that copying some correct vocabulary items may have had a slight impact on otherwise completely incorrect translations.

## 6 Related Work

**Hypothesis Proposal with Language Models** Recent work explores hypothesis proposal to improve language model performance in synthetic tasks, namely ACRE, the original MiniSCAN, ListOps, and versions of the ARC dataset (Qiu et al., 2024; Wang et al., 2024). These methods often rely on domain-specific interpreters or code generation by LMs, akin to our functions domain's ground-truth reranker. We additionally explores probability-based reranking for rule selection across different domains and assesses the accuracy of model-induced rules prior to reranking.

---

[9]This means agreement cannot be computed with the point-biserial correlation.

Language models have also been used to automate hypothesis discovery as an end in itself, to discover distributional differences in text (Zhong et al., 2022; 2023). In this case, the hypothesis proposer LM is paired with a validator trained to filter out irrelevant hypotheses. We similarly find that hypotheses generated by language models themselves may not be very accurate inherently.

**Abductive Reasoning in Language Models** Datasets for abductive commonsense reasoning (Wang et al., 2019; Bhagavatula et al., 2020; Zhang et al., 2020) and logical reasoning (Sinha et al., 2019; Yang et al., 2024) have been proposed. Bhagavatula et al. (2020) focuses on the ability for a model to select the more plausible hypothesis from pairs of hypotheses, while Yang et al. (2024) focuses on proposing natural language rules and evaluating them against ground-truth human rules. Zhao et al. (2023) use contrastive explanations to tune a model to recognize fluent ones. Comparatively, we do not focus on commonsense reasoning, and examine relationships between reasoning types on the same task.

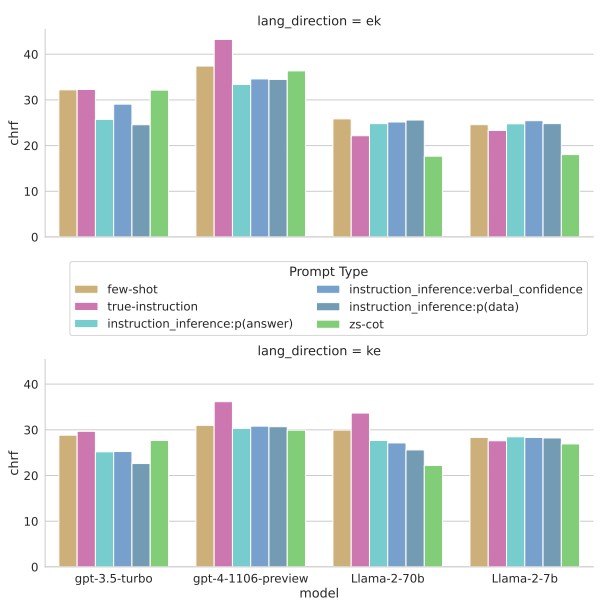

Figure 4: chrF scores for Kalamang under different methods, in English to Kalamang direction (top row) and Kalamang to English direction (bottom row)

**Program Synthesis and Library Learning** The general approach we outline for abductive reasoning can be considered a soft form of program synthesis with language models. Library learning, in which a set of reusable tools is learned from examples, is related (Ellis et al., 2023). A similar approach to ours using LLMs is proposed by Zhu et al. (2023), who use a similar two-step process to learn a library of rules for an arithmetic domain as well as a previous synthetic dataset (Sinha et al., 2019). We find similar gains on synthetic domains, and further examine challenges in applying abductive reasoning in complex domains.

**LM-generated Instructions** Instruction backtranslation is a related concept (Honovich et al., 2023; Li et al., 2024), in which an LM generates instructions for textual data. However, it differs in that the proposed instructions are not directly used at the same time that they are proposed, and it does not focus on generating rules.

## 7 Conclusion

We have examined the interplay between deductive, inductive, and abductive reasoning in LMs through the tasks of hypothesis proposal, in-context learning, and self-generated instruction following. Across three domains (linear function learning, artificial language translation, and Kalamang translation), we show that instruction inference is able to improve over few-shot prompting in simple synthetic domains, but that the relationship between these types of reasoning is complex, and they may not work together as expected presently when models are solving complex tasks. As abductive reasoning seems to be a relatively weaker capability in current language models as compared to instruction following, future work could develop more advanced mechanisms for natural-language hypothesis verification and correction. The use of hypothesis proposal during training remains underexplored, and joint training of models on question answering and hypothesis proposal with enforced consistency may help models display more consistent behaviour. Enhancements in these areas could accelerate progress towards models capable of autonomous learning and self-improvement.

**Acknowledgements**

Thank you to Kiril Gashteovski, Ekin Akyürek, Shuyan Zhou, and Linlu Qiu for helpful discussions on this project. Additional thanks to David Mortensen for suggestions on grammar features and linguistic consultation. Thank you to NEC Labs Europe for supporting this project through the Student Research Fellowship program. We also acknowledge the support of the National Sciences and Engineering Research Council of Canada (NSERC) through a postgraduate fellowship given to EL (PGSD). We also appreciate support from Intel and the National Science Foundation through grants CCF-2217064 and IIS-2238240.

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

# Appendix

## A   Model Inference Settings

Generation settings, as well as other details, used in the three domains are detailed here. Each model was tested 6 times for each setting (`few-shot, zs-cot...`) at different temperatures, and the aggregate results are shown in figures and tables throughout the paper. The exact model versions used for `gpt` models were `gpt-3.5-turbo-0613` and `gpt-4-1106-preview`.

**Functions** When answering questions in the base task, `gpt` models were tested 3 times each at temperature $T = \{0, 1\}$ (note that $T = 0$ is nondeterministic in `gpt` models because of hardware). No max number of tokens was set for the generation. For `llama` models, $T = \{0.1, 1\}$ was used, also with no max number of tokens.

When generating hypotheses, we always used the same model as performed the base task. That is to say, `gpt-3.5-turbo` would generate hypotheses used by `gpt-3.5-turbo`, and so on. Hypotheses were always generated with a temperature above 0 to encourage generation of diverse hypotheses. For `gpt` models, hypotheses were generated with $T = 1$ and for `llama` models, with $T = 1.0625$. The specific value for `llama` models was because `llama` would usually generate the same hypothesis 5 times at $T = 1$, but higher values greatly increased the number of nonsensical and badly formatted hypotheses.

When self-evaluating hypotheses, the verbalized confidence score was generated at $T = 0$. When using the log-probabilities from `text-davinci-002` to rerank hypotheses, the model was also set to $T = 0$.

**Colours** As in the functions domain, `gpt` models and `llama` models were respectively tested 3 times each at temperature $T = \{0, 1\}$ and $T = \{0.1, 1\}$ with no max number of tokens.

When generating hypotheses, $T = 1$ was used for all models.

When self-evaluating hypotheses, settings were the same as in the functions domain.

**Kalamang** For the base translation task, the same settings were used for generation as in Tanzer et al. (2024). Due to cost constraints, we ran only once in each translation direction on each setting with $T = 0.05$.

Vocabulary hypotheses for all models were generated with $T = 1$. Grammar feature hypotheses were generated independently of translations, and the first non-null hypothesis was chosen due to cost constraints. $T = 0.7$ was used for `gpt` grammar hypotheses, and $T = 1$ was used for `llama` grammar hypotheses.

When self-evaluating hypotheses, $T = 0$ was once again used for all models.

# B  Prompts for Linear Functions Domain

Table 5: Prompts for the functions domain. Newlines are depicted visually for ease of reading. Variables that are substituted depending on the question are marked like {this}. The wording for the "prompt with self-induced hypothesis" and "zero-shot chain-of-thought prompt" were slightly changed from the few-shot examples prompt because models were sometimes confused by long-winded hypotheses, and responded with a long-winded answer in return, causing failures to parse their answers. In comparison, models mostly returned outputs in correct formats in other settings.

| Prompt Type | Usage | Prompt Text |
|---|---|---|
| Base system prompt | For reasoning with in-context examples | You are a problem solving system. Your job is to use the input-output pairs to solve the problem as well as you can. |
| Hypothesis proposal system prompt | For proposing hypotheses based on in-context examples | You are a pattern recognition system. Your job is to come up with a function that describes the data as well as you can. |
| Instruction following system prompt | For applying a proposed hypothesis or ground-truth hypothesis to the input | You are a problem solving system. Your job is to apply the function to the data in order to produce an answer. |
| Few-shot examples prompt | For reasoning with in-context examples only | Return the output preceded by 'Output:'
Input: {input1}
Output: {output1}

Input: {input2}
Output: {output2}
...
Input: {query input} |
| Prompt with ground-truth hypothesis | Used when prompting the model to directly apply the correct hypothesis to the input. In-context examples are also included. | Function:
{The real function}

Examples:
Input: {input1}
Output: {output1}

Input: {input2}
Output: {output2}
...
Return the output preceded by 'Output:'
Input: {query input} |
| Prompt for hypothesis induction | Used to have the model propose a single hypothesis for the function based on in-context examples | Write the function that captures the relationship between inputs and outputs.
You should write it in the form $y = ax^0 + bx^1$.
Input: {input1}
Output: {output1}

Input: {input2}
Output: {output2}
...
Function (please write explicitly in the exact form"
'Output: $y = ax^0 + bx^1$)': |

Table 5: Prompts for the functions domain. Newlines are depicted visually for ease of reading. Variables that are substituted depending on the question are marked like {this}. The wording for the "prompt with self-induced hypothesis" and "zero-shot chain-of-thought prompt" were slightly changed from the few-shot examples prompt because models were sometimes confused by long-winded hypotheses, and responded with a long-winded answer in return, causing failures to parse their answers. In comparison, models mostly returned outputs in correct formats in other settings.

| | | |
|---|---|---|
| Prompt with a self-induced hypothesis | Used similarly to the "prompt with ground-truth hypothesis", except with a self-generated hypothesis. The wording is slightly changed. | ```Use this function to apply to the input example to get the correct output.

{ model's hypothesis }

However, just write the output like what's shown in these examples.
Input: {input1}
Output: {output1}
Input: {input2}
Output: {output2}
...
Return the output preceded by 'Output:'
Input: {query input}``` |
| Prompt for zero-shot chain-of-thought | Used to encourage the model to generate a chain of thought. | ```Return the output preceded by 'Final Output:'
Input: {input1}
Output: {output1}

Input: {input2}
Output: {output2}
...
Let's think step by step about what the function could be. Remember to write down 'Final Output:' before your final answer.
Input: {query input}``` |
| Prompt for hypothesis probability estimate | Used to prompt a language model directly for reranking hypotheses | ```How likely is this hypothesis about the function to be true given the data?

Examples:
Input: {input1}
Output: {output1}

Input: {input2}
Output: {output2}
...
Function explanation: {model's hypothesis}

Please give a probability between 0 and 1 inclusive, and only answer with a number.

Probability:``` |
| Prompt for data logprobs given hyp estimate | Used to rerank hypotheses based on logprobs. | ```These are examples of applying this function:
{model's hypothesis}

Examples:
Input: {input1}
Output: {output1}

Input: {input2}
Output: {output2}
...``` |

## C   Prediction Fit of Other Models on Linear Functions

Figure 5 shows predictions made by each model when using a few-shot prompt, true instruction, or self-induced instruction.

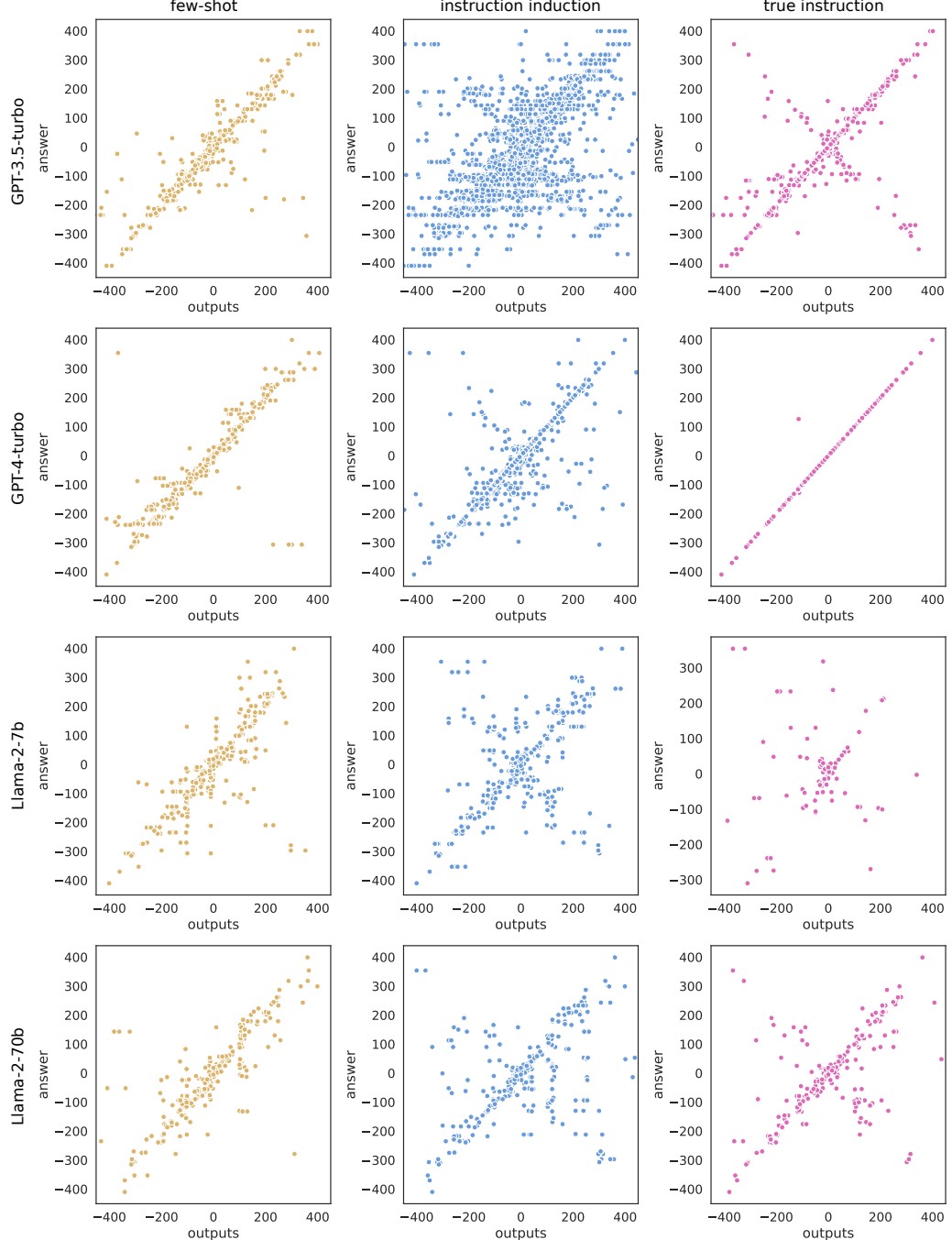

Figure 5: Model predictions plotted against true function output for all models. Range is restricted to the [-400, 400] range for visualization purposes, although there are large outlier values for all models.

# D  Predicted Coefficients Compared to Real Coefficients in Linear Functions

Figure 6 plots model hypotheses about the coefficients of $x^0$ and $x^1$ in a linear function against the actual coefficients

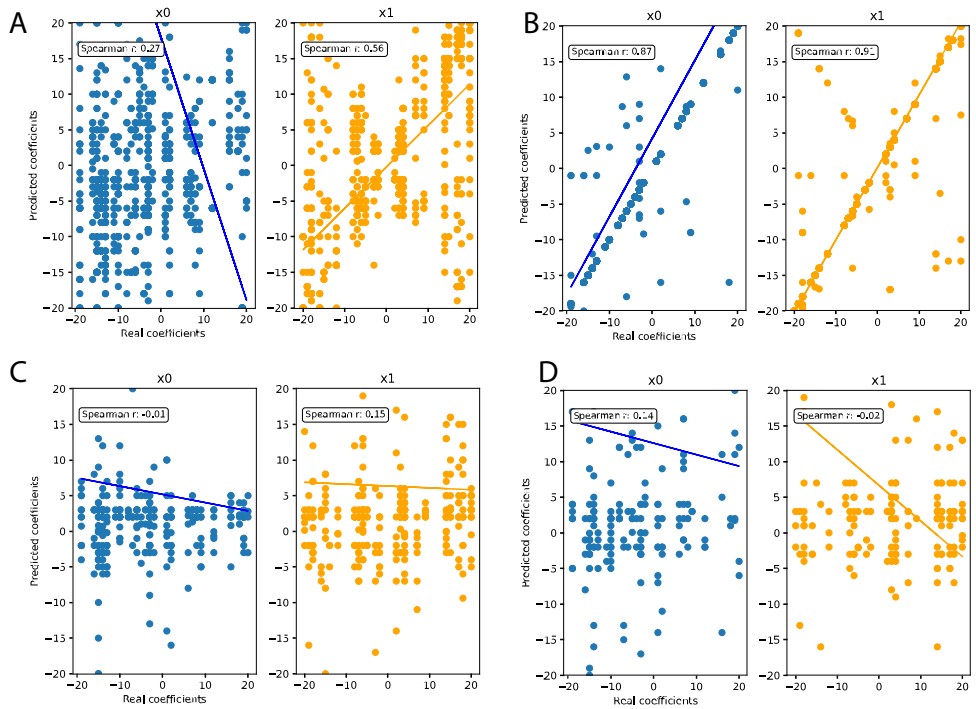

Figure 6: Model predictions plotted against true function output for all models. Range is restricted to the [-400, 400] range for visualization purposes, although there are large outlier values for all models.

# E  Grammar and Details of Colours Domain

The rules of the colours domain are expressed through production rules below. Models were found to respond better to verbal instructions than formal ones in initial testing (e.g. a verbal statement "repeat twice" rather than $[[x]]\texttt{bluf} \rightarrow [[x]][[x]]$)

Listing 1: Grammar of the colours language, as presented to LMs.

```
lug -> blue
dax -> green
wif -> red
zup -> yellow
bluf -> repeat the last action twice
walm -> repeat the last action three times
```

Training and test data was automatically generated by generating sentences of up to 5 nonce words on the source side, with shorter sentences being more likely (the respective probabilities for sentence lengths from 1 to 5 are [0.4, 0.3, 0.15, 0.1, 0.05]). Each colour could also be repeated with the repeat actions, and whether repeat nonce words were inserted was also random, though skewed toward no repetition (the probabilities were respectively [0.8,

0.1, 0.1] for no repeats, one repeat, or two repeats). A repeat term never followed another repeat term, and the same colour word never appeared twice consecutively to make it easier to learn the repeat terms.

Additionally, we provided a fixed set of few-shot examples that covered all the nonce terms:

Listing 2: Few-shot examples in the colours translation task

```
Input: lug dax
Output: blue green

Input: wif zup
Output: red yellow

Input: lug bluf
Output: blue blue

Input: wif walm
Output: red red red

Input: lug walm dax bluf
Output: blue blue blue green green
```

# F   Prompts for Colours Domain

Table 6: Prompts for the colours domain. Newlines are depicted visually for ease of reading. Variables that are substituted depending on the question are marked like {this}. The "prompt with a self-induced hypothesis" was slightly modified from the base prompt in order to encourage models to follow the correct formatting, while the "prompt for zero-shot chain-of-thought" was slightly modified to encourage models to generate a concrete chain of thought, and also to follow the correct formatting.

| Prompt Type | Usage | Prompt Text |
|---|---|---|
| Base system prompt | For reasoning with in-context examples | You are a problem solving system. Your job is to use the input-output pairs to solve the problem as well as you can.'' |
| Hypothesis proposal system prompt | For proposing hypotheses based on in-context examples | You are a rule induction system. Your job is to figure out the rules underlying a problem and report on them. Use the examples to guide your thinking. |
| Instruction following system prompt | For applying a proposed hypothesis or ground-truth hypothesis to the input | You are a parser. Carefully use the grammar to parse inputs to determine the correct output. |
| Few-shot examples prompt | For reasoning with in-context examples only | Return the output preceded by 'Output:'
Input: {input1}
Output: {output1}

Input: {input2}
Output: {output2}
...
Input: {query input}
Remember to start your answer with 'Output:' |

Table 6: Prompts for the colours domain. Newlines are depicted visually for ease of reading. Variables that are substituted depending on the question are marked like {this}. The "prompt with a self-induced hypothesis" was slightly modified from the base prompt in order to encourage models to follow the correct formatting, while the "prompt for zero-shot chain-of-thought" was slightly modified to encourage models to generate a concrete chain of thought, and also to follow the correct formatting.

| | | |
|---|---|---|
| Prompt with ground-truth hypothesis | Used when prompting the model to directly apply the correct hypothesis to the input. In-context examples are also included. | Use this grammar to parse the input example
to get the correct output.
Grammar:
{colours domain grammar}

Examples:
Input: {input1}
Output: {output1}

Input: {input2}
Output: {output2}
...
Return the output preceded by 'Output:'
Input: {query input} |
| Prompt for hypothesis induction | Used to have the model propose a single hypothesis for the translation of a word. | The below examples contain the nonce word {word}.

Using the examples, deduce what {word} means.
Input: {input1}
Output: {output1}

Input: {input2}
Output: {output2}
...
Write your answer like this: word -> meaning.
Meaning can be a word or a general rule dependent on the context.
Rule: |
| Prompt with a self-induced hypothesis | Used similarly to the "prompt with ground-truth hypothesis", except with a self-generated hypothesis. The wording is slightly changed. | Use this grammar to parse the input example
to get the correct output.


{ model's hypothesis grammar }

However, just write the output like
what's shown in these examples.
Input: {input1}
Output: {output1}
Input: {input2}
Output: {output2}
...
Return the output preceded by 'Output:'
Input: {query input} |

Table 6: Prompts for the colours domain. Newlines are depicted visually for ease of reading. Variables that are substituted depending on the question are marked like {this}. The "prompt with a self-induced hypothesis" was slightly modified from the base prompt in order to encourage models to follow the correct formatting, while the "prompt for zero-shot chain-of-thought" was slightly modified to encourage models to generate a concrete chain of thought, and also to follow the correct formatting.

| | | |
|---|---|---|
| Prompt for zero-shot chain-of-thought | Used to encourage the model to generate a chain of thought. | Return the output preceded by 'Final Output:'
Input: {input1}
Output: {output1}

Input: {input2}
Output: {output2}
...
Let's think step by step about what the translation could be.
Work through your answer step by step and show your work.
Remember to write down 'Final Output:'
before your final answer.
Input: {query input} |
| Prompt for hypothesis probability estimate | Used to prompt a language model directly for reranking hypotheses | These are examples of the
translation of the word {word}.

Examples:
Input: {input1}
Output: {output1}

Input: {input2}
Output: {output2}
...
Given these examples, how likely is
this hypothesis about the meaning of {word}?
{model's word hypothesis}

Please give a probability between 0 and 1 inclusive,
and only answer with a number.

Probability: |
| Prompt for data logprobs given hyp estimate | Used to rerank hypotheses based on logprobs. | These are examples of applying this function:
{model's word hypothesis}

Examples:
Input: {input1}
Output: {output1}

Input: {input2}
Output: {output2}
... |

## G chrF for Colours Domain

Figure 7 depicts the mean chrF score over 6 trials for each model in each setting. Reranking methods are averaged for `instruction-induction`.

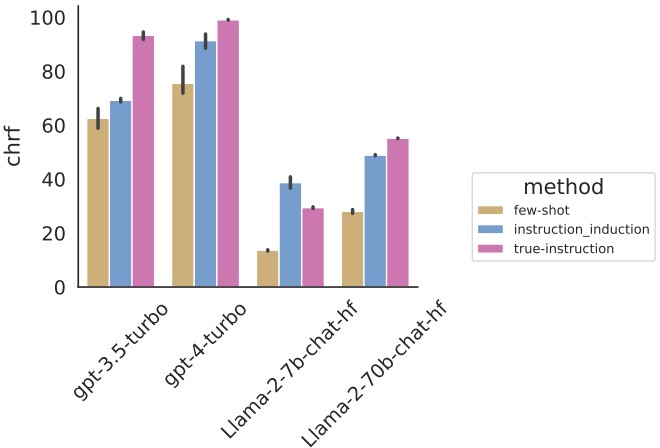

Figure 7: chrF scores for the colours domain under different methods. All reranking methods are averaged.

## H Vocabulary Induction Accuracy for Colours Domain

We show in Table 7 the mean accuracy for each colour term for `gpt` models, computed over all selected (highest-ranked) word hypotheses. Point-biserial correlation between hypothesis correctness (in the `instruction-induction` case) and few-shot correctness is also shown. P-values are corrected with FDR in the trials for each model. We performed the correction per model rather than across all models because we were separately interested in the behaviour of each model. We parsed the production rule generated by the model in order to determine hypothesis correctness.

Table 7: Vocabulary induction correctness and relation to few-shot translation correctness for gpt models. Significant correlations after correction for multiple comparisons are bolded.

| Model | Word | method | Mean Correctness | Correlation | p-value | p-value corrected |
|---|---|---|---|---|---|---|
| gpt-3.5-turbo | lug | verbal_conf | 0.60 | -0.037 | 0.30 | 0.53 |
| gpt-3.5-turbo | dax | verbal_conf | 0.23 | 0.067 | 0.064 | 0.30 |
| gpt-3.5-turbo | wif | verbal_conf | 0.33 | 0.036 | 0.65 | 0.82 |
| gpt-3.5-turbo | zup | verbal_conf | 0.61 | 0.15 | 0.029 | 0.20 |
| gpt-3.5-turbo | bluf | verbal_conf | 0 | – | – | – |
| gpt-3.5-turbo | walm | verbal_conf | 0 | – | – | – |
| gpt-4-turbo | lug | verbal_conf | 0.88 | 0.044 | 0.17 | 3.58e-1 |
| gpt-4-turbo | dax | verbal_conf | 0.92 | -0.042 | 0.19 | 3.58e-1 |
| gpt-4-turbo | wif | verbal_conf | 0.86 | 0.0445 | 0.57 | 7.32e-1 |
| gpt-4-turbo | zup | verbal_conf | 0.87 | 0.058 | 0.42 | 5.75e-1 |
| gpt-4-turbo | bluf | verbal_conf | 0.15 | 0.13 | 0.26 | 3.83e-1 |
| gpt-4-turbo | walm | verbal_conf | 0.17 | 0.049 | 0.69 | 7.55e-1 |
| gpt-3.5-turbo | lug | p_data | 0.29 | 0.059 | 0.098 | 0.30 |
| gpt-3.5-turbo | dax | p_data | 0.10 | 0.010 | 0.78 | 0.84 |
| **gpt-3.5-turbo** | **wif** | **p_data** | **0.20** | **0.24** | **0.0016** | **0.023** |
| gpt-3.5-turbo | zup | p_data | 0.4 | -0.11 | 0.10 | 0.30 |
| gpt-3.5-turbo | bluf | p_data | 0.077 | -0.17 | 0.14 | 0.33 |
| gpt-3.5-turbo | walm | p_data | 0 | – | – | – |
| gpt-4-turbo | lug | p_data | 0.73 | -0.080 | 0.012 | 5.31e-2 |
| gpt-4-turbo | dax | p_data | 0.73 | -0.041 | 0.20 | 3.58e-1 |
| gpt-4-turbo | wif | p_data | 0.77 | -0.026 | 0.74 | 7.55e-1 |
| gpt-4-turbo | zup | p_data | 0.76 | 0.022 | 0.76 | 7.55e-1 |
| gpt-4-turbo | bluf | p_data | 0.10 | -0.14 | 0.22 | 3.58e-1 |
| gpt-4-turbo | walm | p_data | 0.14 | -0.26 | 0.027 | 9.89e-2 |
| gpt-3.5-turbo | lug | p_answer | 0.44 | -0.013 | 0.72 | 0.84 |
| gpt-3.5-turbo | dax | p_answer | 0.19 | 0.0040 | 0.91 | 0.91 |
| gpt-3.5-turbo | wif | p_answer | 0.35 | -0.074 | 0.34 | 0.53 |
| gpt-3.5-turbo | zup | p_answer | 0.59 | -0.088 | 0.21 | 0.43 |
| gpt-3.5-turbo | bluf | p_answer | 0.051 | -0.055 | 0.63 | 0.82 |
| gpt-3.5-turbo | walm | p_answer | 0 | – | – | – |
| gpt-4-turbo | lug | p_answer | 0.70 | -0.066 | 0.038 | 1.15e-1 |
| **gpt-4-turbo** | **dax** | **p_answer** | **0.71** | **0.19** | **3.67e-9** | **6.61e-8** |
| **gpt-4-turbo** | **wif** | **p_answer** | **0.82** | **-0.25** | **0.0013** | **1.13e-2** |
| gpt-4-turbo | zup | p_answer | 0.68 | -0.14 | 0.045 | 1.15e-1 |
| gpt-4-turbo | bluf | p_answer | 0.23 | -0.29 | 0.0089 | 5.31e-2 |
| gpt-4-turbo | walm | p_answer | 0.057 | -0.058 | 0.63 | 7.55e-1 |

## I  MTOB Experimental Settings

We generally used the same experimental setup as in `mtob`. For the `few-shot` condition, we used two reference sentences for each word on the source side, selected via longest subsequence. For the `true-instruction` setting, we used reference sentences, in addition to the wordlist and true grammar sketch. For each word in the source sentence, the most similar word from the wordlist was also retrieved based on the longest common substring. In the `instruction-inference` settings, the model's self-induced grammar sketch was substituted for the true grammar sketch, and the model was also prompted to first create hypotheses about the translation of each word in the source sentence.

At the time that we conducted experiments, the Kalamang to English training set was not available, so we created this training set by reversing the source and target in the English to Kalamang training set. This may have slightly harmed translations in this direction, as this caused the training set to be different from the one reported in the benchmark results.

## J  MTOB Prompts

Table 8: Prompts for Kalamang translation with MTOB. We follow the same prompt formatting as in Tanzer et al. (2024), with the exception of replacing retrieved grammar book passages with the grammar sketch in Appendix K. Newlines are depicted visually for ease of reading. Variables that are substituted depending on the question are marked like {this}. The English to Kalamang direction is depicted in the examples. ' ' ' ' indicates that the prompt is the same as in Appendix F.

| Prompt Type | Usage | Prompt Text |
|---|---|---|
| Base system prompt | For reasoning with in-context examples | ' ' ' ' |
| Hypothesis proposal system prompt (Grammar) | For proposing grammar feature hypotheses based on in-context examples | ' ' ' ' |
| Hypothesis proposal system prompt (Vocab) | For proposing vocabulary mappings based on in-context examples | ' ' ' ' |
| Instruction following system prompt | For applying a proposed hypothesis or ground-truth hypothesis to the input | ' ' ' ' |

Table 8: Prompts for Kalamang translation with MTOB. We follow the same prompt formatting as in Tanzer et al. (2024), with the exception of replacing retrieved grammar book passages with the grammar sketch in Appendix K. Newlines are depicted visually for ease of reading. Variables that are substituted depending on the question are marked like {this}. The English to Kalamang direction is depicted in the examples. '''' indicates that the prompt is the same as in Appendix F.

| Few-shot examples prompt | For reasoning with in-context examples only | ```Human: Kalamang is a language spoken on the Karas Islands in West Papua. Translate the following sentence from English to Kalamang: {query input}``` ```To help with the translation, here is a translated sentence with words similar to '{word}' in a list of translated Kalamang-English reference sentences: English sentence: {eng sentence} Kalamang translation: {kgv sentence} <Repeated for all words> Now write the translation. English: {query input} Kalamang translation:``` |
|---|---|---|
| Prompt with ground-truth hypothesis | Used when prompting the model to directly apply the correct hypothesis to the input. In-context examples are also included. | ```Human: Kalamang is a language spoken on the Karas Islands in West Papua. Translate the following sentence from English to Kalamang: {query input}``` ```To help with the translation, here is one of the closest entries to '{word}' in the Kalamang=English bilingual dictionary: English word or phrase: '{word}' Kalamang translation: '{translation}' <Repeated for all words>``` ```To help with the translation, here is a translated sentence with words similar to '{word}' in a list of translated Kalamang-English reference sentences: English sentence: {eng sentence} Kalamang translation: {kgv sentence} <Repeated for all words>``` ```To help with the translation, here's a grammar sketch of Kalamang: {grammar sketch}``` ```Now write the translation. English: {query input} Kalamang translation:``` |
| Prompt for hypothesis induction | Used to have the model propose a single hypothesis for the translation of a word. | ```The following sentences contain the word '{word}' in English. Examples: English sentence: {eng sentence} Kalamang translation: {kgv sentence} ...``` ```What is the Kalamang translation of the word '{word}'? Write your answer like this: {English word} -> {Kalamang translation}.``` |

Table 8: Prompts for Kalamang translation with MTOB. We follow the same prompt formatting as in Tanzer et al. (2024), with the exception of replacing retrieved grammar book passages with the grammar sketch in Appendix K. Newlines are depicted visually for ease of reading. Variables that are substituted depending on the question are marked like {this}. The English to Kalamang direction is depicted in the examples. ''' indicates that the prompt is the same as in Appendix F.

| Prompt with a self-induced hypothesis | Used similarly to the "prompt with ground-truth hypothesis", except with a self-generated hypothesis. The wording is slightly changed. | Human: Kalamang is a language spoken on the Karas Islands in West Papua. Translate the following sentence from English to Kalamang: {query input}

To help with the translation, here is one of the closest entries to '{word}' in the Kalamang=English bilingual dictionary: English word or phrase: '{word}' Kalamang translation: '{hypothesis translation}' <Repeated for all words>

To help with the translation, here is a translated sentence with words similar to '{word}' in a list of translated Kalamang-English reference sentences:

English sentence: {eng sentence} Kalamang translation: {kgv sentence} <Repeated for all words>

To help with the translation, here's a grammar sketch of Kalamang: {hypothesis grammar sketch}

Now write the translation. English: {query input} Kalamang translation: |
| Prompt for zero-shot chain-of-thought | Used to encourage the model to generate a chain of thought. | Human: Kalamang is a language spoken on the Karas Islands in West Papua. Translate the following sentence from English to Kalamang: {query input}

To help with the translation, here is a translated sentence with words similar to '{word}' in a list of translated Kalamang-English reference sentences:

English sentence: {eng sentence} Kalamang translation: {kgv sentence} <Repeated for all words> Let's think step by step about what the translation could be. Now write the translation. English: {query input} Kalamang translation: |

Table 8: Prompts for Kalamang translation with MTOB. We follow the same prompt formatting as in Tanzer et al. (2024), with the exception of replacing retrieved grammar book passages with the grammar sketch in Appendix K. Newlines are depicted visually for ease of reading. Variables that are substituted depending on the question are marked like {this}. The English to Kalamang direction is depicted in the examples. ''' indicates that the prompt is the same as in Appendix F.

| Prompt for hypothesis probability estimate | Used to prompt a language model directly for reranking hypotheses | These are examples of the translation of the word {word}.

Examples:
{example translation pairs}
...
Given these examples, how likely is this hypothesis about the meaning of {word}?
{model's word hypothesis}

Please give a probability between 0 and 1 inclusive, and only answer with a number.

Probability: |
|---|---|---|
| Prompt for data logprobs given hyp estimate | Used to rerank hypotheses based on logprobs. | These are examples of sentences that contain the word {word}:
This is the translation of the word:
{model's word hypothesis}

Examples:
{example translation pairs} |

## K   Kalamang Grammar Sketch

Listing 3: Grammar sketch for Kalamang, compiled from WALS and Grambank features.

```
=== Start of grammar sketch ===
Basic Word Order: SV (Subject-Verb), OV (Object-Verb)
Noun Phrase Construction: Postpositional, Genitive-Noun, Noun-Adjective, Noun-
    Demonstrative, Noun-Num, Possessed-Possessor
Articles: No definite/specific or indefinite articles
Morphological Marking: No productive singular, dual, or plural marking on nouns
    ; Possession marked by suffix on the possessed noun; Tense marked by
    auxiliary particle
Syntactic Alignment: Accusative
Negation: Standard negation marked clause-finally; Distinct imperative negation
Reduplication: Both verbs and nouns can be reduplicated
=== End of grammar sketch ===
```

## L   Predicted Kalamang Grammar Sketches and Accuracy

Listing 4: Grammar sketch for Kalamang, generated by GPT-3.5-turbo. Each feature was predicted individually based on randomly retrieved examples from the training data.

```
=== Start of grammar sketch ===
Basic Word Order: SV (Subject-Verb), OV (Object-Verb)
Noun Phrase Construction: Adposition-Noun Phrase, Genitive-Noun, Adjective-Noun
    , Noun-Demonstrative, Noun-Num, Possessor-Possessed
Articles: Unsure if there are definite/specific or indefinite articles
Morphological Marking: Unsure if singular, dual, or plural marking on nouns;
    Unsure if possession marked by suffix on the possessed noun; Unsure if
    tense marked by auxiliary particle
```

```
Syntactic Alignment: Unsure
Negation: Standard negation marked clause-initially; Unsure if distinct
    imperative negation
Reduplication: Unsure if verbs and nouns can be reduplicated
=== End of grammar sketch ===
```

Listing 5: Grammar sketch for Kalamang, generated by GPT-4-turbo. Each feature was predicted individually based on randomly retrieved examples from the training data.

```
=== Start of grammar sketch ===
Basic Word Order: SV (Subject-Verb), VO (Verb-Object)
Noun Phrase Construction: Unsure on order of adposition and noun phrase,
    Genitive-Noun, Adjective-Noun, Unsure on order of noun and demonstrative,
    Num-Noun, Possessor-Possessed
Articles: No definite/specific or indefinite articles
Morphological Marking: Unsure if productive singular, dual, or plural marking
    on nouns; Unsure if Possession marked by suffix on the possessed noun;
    Unsure if tense marked by auxiliary particle
Syntactic Alignment: Unsure
Negation: Standard negation marked clause-initially; Unsure if distinct
    imperative negation
Reduplication: Unsure if verbs and nouns can be reduplicated
=== End of grammar sketch ===
```

| Model | Grammar feature accuracy |
|---|---|
| GPT-3.5-turbo | 27.78% |
| GPT-4-turbo | 22.22% |
| Llama-2-7B | 0% |
| Llama-2-70B | 0% |

Table 9: Overall grammar sketch accuracy of models. Unsure answers were marked as incorrect.

## M  Vocabulary Induction Accuracy for MTOB

The overall vocabulary induction accuracy for gpt models is shown in Table 10. Accuracy and chrF were calculated overall when exclude morphology is marked No, and morphologically rich words in the dictionary (marked with * or with - to represent suffixes/prefixes) were excluded when this value was marked Yes. For words with multiple translations in the dictionary, the translation was marked correct if it matched any of the possible translations. Words without an entry in the dictionary were skipped. If a null hypothesis such as *I don't know* was proposed, it was marked as incorrect.

Due to hypotheses for this task being more difficult to automatically evaluate, an author also annotated the correctness of results for a single setting (instruction-inference:verbalized confidence) in order to calculate the association between translation quality and hypothesis correctness more accurately. Results can be found in Table 11.

Table 10: Accuracy and segment-level chrF for vocabulary hypotheses in Kalamang translation. If morphology was excluded, all words with a * or - symbol in the translation (corresponding to prefixes/suffixes) were excluded, whereas otherwise we matched whether or not the model's hypothesis began/ended with the correct characters.

| direction | model | rerank method | exclude morphology | acc | chrF |
|---|---|---|---|---|---|
| ek | gpt-4-turbo | p_data_given_hyp_guess | No | 0.1528 | 20.33 |
| ek | gpt-4-turbo | p_data_given_hyp_guess | Yes | 0.0813 | 11.86 |
| ek | gpt-4-turbo | p_answer_given_hyp_logprobs | No | 0.16 | 20.69 |
| ek | gpt-4-turbo | p_answer_given_hyp_logprobs | Yes | 0.08621 | 12.18 |
| ek | gpt-4-turbo | p_data_given_hyp_logprobs | No | 0.166 | 20.79 |
| ek | gpt-4-turbo | p_data_given_hyp_logprobs | Yes | 0.1015 | 13.21 |
| ke | gpt-3.5-turbo | p_answer_given_hyp_logprobs | No | 0.1033 | 15.33 |
| ke | gpt-3.5-turbo | p_answer_given_hyp_logprobs | Yes | 0.8244 | 13.11 |
| ke | gpt-3.5-turbo | p_data_given_hyp_guess | No | 0.11 | 16.42 |
| ke | gpt-3.5-turbo | p_data_given_hyp_guess | Yes | 0.09018 | 14.33 |
| ke | gpt-3.5-turbo | p_data_given_hyp_logprobs | Yes | 0.0738 | 12.34 |
| ke | gpt-3.5-turbo | p_data_given_hyp_logprobs | No | 0.1014 | 15.28 |
| ke | Llama-2-70b | p_data_given_hyp_guess | Yes | 0.1344 | 21.69 |
| ke | Llama-2-70b | p_data_given_hyp_guess | No | 0.155 | 23.62 |
| ek | gpt-3.5-turbo | p_data_given_hyp_guess | Yes | 0.0922 | 15.05 |
| ek | gpt-3.5-turbo | p_data_given_hyp_guess | No | 0.1217 | 19.29 |
| ke | gpt-4-turbo | p_answer_given_hyp_logprobs | Yes | 0.08127 | 13.54 |
| ke | gpt-4-turbo | p_answer_given_hyp_logprobs | No | 0.0979 | 15.25 |
| ke | gpt-4-turbo | p_data_given_hyp_logprobs | Yes | 0.07641 | 12.82 |
| ke | gpt-4-turbo | p_data_given_hyp_logprobs | No | 0.0924 | 14.46 |
| ke | Llama-2-70b | p_answer_given_hyp_logprobs | Yes | 0.1339 | 21.92 |
| ke | Llama-2-70b | p_answer_given_hyp_logprobs | No | 0.1447 | 22.92 |
| ke | Llama-2-70b | p_data_given_hyp_logprobs | Yes | 0.1334 | 21.36 |
| ke | Llama-2-70b | p_data_given_hyp_logprobs | No | 0.1442 | 22.38 |

| Model | Correlation | p-val |
|---|---|---|
| **GPT-3.5-turbo** | **0.17** | **0.0063** |
| GPT-4-turbo | 0.12 | 0.051 |

Table 11: Point-biserial correlation between segment-level chrF of each translation with the correctness of vocabulary hypotheses induced for that question (in the `instruction-inference:verbalized confidence` setting). No correction for multiple comparisons was done because we examined hypotheses at the model level (see Appendix H for how this was done in the colours domain)

## N  Example Kalamang Translations

Table 12 shows example translations made by GPT-3.5-turbo in both directions.

Table 12: Example translations by the GPT-3.5-turbo model in different settings.

| Source | Reference | Translation (few-shot) | Translation (instruction-induction:verbalized confidence) | Translation (true-instruction) |
|---|---|---|---|---|
| Who asked you? kiemara paruo? | Namana kat ger-ket? | Naman mua kiemara kowam? | Naman era? | Naman mua |
| The rope is wrapped around it. | Karek mat kokarap. | Kainasu esun kiwat semda bo dalangdalang. | Sar tok esun et dalangdalang na. | Karek mambon guanggarien ma he me. |
| Before, they said Japan and The Netherlands were at war. | Wiseme mu toni Japangbon Be-ladarbon nausair. | Wiseme mu Ja-faris emun Nabil emun kaden kiesun ning. | Wise me mu salai Japan an Sar Be-ladargara din jam-bat mara. | Tikninda, termus siram panci tubak Beladar tamatko pang. |
| Mu bo Tamisen-gga som dande yecie. | They are going (on a return trip) to bury someone in Antalisa. | They go to Tamisengga and return. | They go to An-talisa, someone died, and they return. | They will return to Tamisengga un-til they bury the person. |
| Suo me kene arkon metko. | At Suo is a kene tree. | I washed my body in the river. | Back me gives birth canoe. | Wash your face near the Narkon tree. |
| An istrat kahendengoa marmar ba temun. | I'm walking on a long wide road. | I put the long stick on the ground. | I have a long dry-ing rack but I am walking. | I walked far down the street to my uncle. |

