# OpenReview forum: "An Incomplete Loop: Instruction Inference, Instruction Following, and In-Context Learning in Language Models"
_colmweb.org/COLM/2024/Conference — COLM_

### Official Review · Reviewer_Ugsb · 2024-05-05

**Rating:** 9
**Confidence:** 4
**Ethics Flag:** 1

**Summary:**

This paper investigates whether LMs possess the types of reasoning necessary to robustly learn to perform and describe a task in-context. Specifically, the authors propose a reasoning loop consisting of deductive, inductive, and abductive reasoning, and then test whether LMs can perform each type of reasoning. Using coefficient prediction, nonce word translation, and machine translation tasks, the authors test the ability of models to follow instructions (deductive reasoning), generalize from few-shot prompts (inductive reasoning), and induce instructions given labeled examples.

The contribution of this paper is clear and original, and it yields a more fine-grained understanding of what kinds of in-context reasoning, exactly, will be more difficult for LLMs to accomplish.

**Questions To Authors:**

Suggestions:
- The Kalamang grammar row of Table 2 looks funny. This task setup is very different from the others, so maybe it makes more sense to simply put this in a separate table where the task setup would be better described using different columns.
- In Figure 3, could you reorder the legend rows to match with the order of the bars? Also, I didn’t notice at first that the y-axis ranges were different across subfigures, which made me think that the Llama scores were better than they actually were. Maybe you could use the same [0, 1] range for all subfigures?
- When linking to RQ1 and RQ2 (as on p.6), it might look better to exclude the colon.
- Inconsistent references to subfigures within Figure 3. First it’s “3A” and “3B”, but in the next paragraph, it’s “3(A)” and “3(B)”.’

Typos:
- Add period to end of Footnotes 4 and 10.
- Why are “few-shot”, “zero-shot”, and “chain-of-thought” in \texttt{} format on p.3 but nowhere else in the paper?
- Add period after “Appendix F” at end of "$\textbf{Hypotheses}$” on p.5.
- Remove space before “Settings” at top of p.4.

**Reasons To Accept:**

A1. The hypothesized reasoning structure is specific, falsifiable, and broken down into testable components.

A2. The results provide evidence that these reasoning mechanisms are separate in LLMs. It is interesting that prediction accuracy is not closely related to the ability to perform instruction induction.

A3. Reproducibility seems good. Detailed experimental setup details (including all prompts and hyperparameters) are provided in the Appendix.

A4. The writing style and figures are clear. (The paper is also just a joy to read. I especially appreciate the frequent pre-21st-century citations.)

**Reasons To Reject:**

R1. Table 3 is opaque. What were the criteria for deciding whether a cell got a checkmark, partial check, or X? Were these qualitative judgments, or based on statistical significance tests?

R2. The Kalamang translation results are weak, and don’t seem to provide especially strong evidence for either RQ. That said, this is a very difficult task, and I appreciate its inclusion; I therefore don’t think this is a significant weakness.

---

> ### Author Rebuttal · Authors · 2024-05-30
>
> We’re very glad to hear that you enjoyed the paper, and thank you for your detailed suggestions! We have incorporated your suggestions into the current draft, and please see below for responses to your comments:
>
> > R1. Table 3 is opaque. What were the criteria for deciding whether a cell got a checkmark, partial check, or X? Were these qualitative judgments, or based on statistical significance tests?
>
> This was qualitative, based on the actual accuracy for “Deductive/abductive/hypothesis proposal works”, and the checkmark/partial check, or X was based on whether respectively ground truth rules and instruction inference improved performance for the first two columns, and whether the hypotheses were actually mostly correct for the third column.
> The purpose of Table 3 was mainly to summarize the takeaways for readers, rather than to provide detailed results. We will clarify the meanings of the symbols in the caption for the camera-ready version.
>
> > R2. The Kalamang translation results are weak, and don’t seem to provide especially strong evidence for either RQ. That said, this is a very difficult task, and I appreciate its inclusion; I therefore don’t think this is a significant weakness.
>
> Thank you for appreciating the inclusion of Kalamang, we are aware that the results are rather mixed in terms of not following the trends in the functions and colours domains, but we thought it was important to include a real-world task which was more difficult, especially since similar papers have also been tested on more synthetic domains. It is possible that with more work in the low-resource translation domain, instruction inference may work better and similar trends may be seen in this domain as well, but we leave this to future work.
>
>
> > Typos/suggestions
>
> Thank you for your detailed reading in making these suggestions and finding all these typos! We have corrected all typos in our draft. In addition, we will also address the suggestions in the camera-ready draft.

---

> > ### Comment · Reviewer_Ugsb · 2024-06-03
> > **Thank you**
> >
> > Thanks for your response. Since I already gave a very high score and these weaknesses are mostly going to be the same (with clarification on R1), and taking the other reviews into account, I'm choosing keeping my score the same. I look forward to reading the final version of this work!

---

### Official Review · Reviewer_zPDQ · 2024-05-11

**Rating:** 6
**Confidence:** 4
**Ethics Flag:** 1

**Summary:**

This paper evaluates the performance of three styles of prompting, namely, instruction following, few-shot learning, and instruction inference, on four tasks.
The main findings are:
- instruction inference, which asks an LLM to generate hypotheses given examples and apply the best hypothesis to the input, is effective on simple tasks but not on complex tasks (low-resource translation and grammatical knowledge acquisision), and,
- the relationship among the three styles of prompting is not observed.

**Reasons To Accept:**

The experimental results shown in the paper are useful and meaningful to see which style of prompting is effective on which tasks.
However, it is difficult to obtain any general insights from the results and to generalize these results to other tasks.

**Reasons To Reject:**

As described above, the experimental results are meaningful but no generalized knowledge is obtained from these results.
What we can know is that the effectiveness of different prompting styles is mixed, and this is kind of expectable.

The main focus of the paper is on three styles of prompting, i.e., instruction following, few-shot learning, and instruction inference, rather than generic logical inference like deductive, inductive, and abductive reasoning.
The paper describes these three methods as deductive, inductive, and abductive learning, while they are just a specific instance of these logical inference patterns.
It sounds overgeneralization to use these logical terms in the title and the main description of the work.
In other words, the paper does not provide any generic observations for deductive/inductive/abductive learning ability of LLMs.

---

> ### Author Rebuttal · Authors · 2024-05-30
>
> Thank you for your comments, we’re glad that you see potential for future research! Please see responses below:
>
> > As described above, the experimental results are meaningful but no generalized knowledge is obtained from these results. What we can know is that the effectiveness of different prompting styles is mixed, and this is kind of expectable.
>
> We agree with the need to further investigate in more domains and examine the broader applicability of the findings. However, technically a lot of results related to prompting can be summarized as “the effectiveness of different prompting styles is mixed”. For instance, chain-of-thought reasoning can be summarized this way, since CoT sometimes degrades performance as well. However, knowing this would still (1) provide an easy and practical way for people to get more performance out of their models (2) reveal future avenues for investigating the inner workings of LMs. In our case, instruction inference can be used relatively easily to guide LMs, at least on simple tasks. Furthermore, the dissociation between hypothesis proposal and in-context reasoning is interesting, and may merit further investigation. For instance, perhaps by training to make proposed hypotheses amenable with LM outputs without the ground truth label, we can increase reliability of models.
>
> More broadly, we would like to point out that many highly impactful papers identify problems without proposing a solution in the same paper (such as the original papers on adversarial examples in NLP/vision, which are arguably still not fixed). However, knowing that this problem exists can guide the community in best practices for using and developing models.
>
> > The main focus of the paper is on three styles of prompting, i.e., instruction following, few-shot learning, and instruction inference, rather than generic logical inference like deductive, inductive, and abductive reasoning. The paper describes these three methods as deductive, inductive, and abductive learning, while they are just a specific instance [...]
>
>
> Thank you for bringing this up, we plan to clarify the terminology more in the final version. We can clarify more in Section 2 that we are not making claims about these capabilities in general, but rather on common instantiations of these types of reasoning in LMs (hence the use of “learning” rather than “reasoning” in the title).

---

> > ### Comment · Reviewer_zPDQ · 2024-06-03
> > **Thanks for the responses.**
> >
> > I totally agree that it is meaningful to point out problems and you don't have to present complete solutions.
> > This is still a weakness (given findings are not extremely surprising compared to e.g. adversarial examples), but understandable.
> > I still think using overgeneralized terms is not a good practice (although we see many such papers recently).
> > I prefer the title like "Investing instruction following, few-shot learning, and instruction inference..." that clearly describes what is done in the paper.

---

### Official Review · Reviewer_z3T2 · 2024-05-11

**Rating:** 6
**Confidence:** 2
**Ethics Flag:** 1

**Summary:**

This paper examines the relationship between instruction following, few-shot prompting, and instruction inference, linking them to deductive reasoning, inductive reasoning, and abductive reasoning, respectively. It explores these concepts across three domains: linear function learning, artificial language translation, and Kalamang translation. The findings suggest that instruction inference can surpass few-shot prompting in simpler tasks; however, its efficacy diminishes in more complex tasks such as low-resource machine translation. Additionally, the paper observes that a model's capability to generate hypotheses is not correlated with its ability to learn from few-shot examples.

**Reasons To Accept:**

The study conducts an extensive examination of the interplay between three prevalent reasoning paradigms, providing empirical insights that could be valuable for future research. This encourages the development of innovative methods for hypothesis verification and correction in natural language processing.

**Reasons To Reject:**

While the paper concludes that the ability to generate hypotheses is unrelated to learning from few-shot examples, it fails to suggest future directions or solutions to address this gap. This omission may limit the paper's impact on advancing the field's understanding of integrating these capabilities.

---

> ### Author Rebuttal · Authors · 2024-05-30
>
> Thanks for your comments! See below for response:
>
> > While the paper concludes that the ability to generate hypotheses is unrelated to learning from few-shot examples, it fails to suggest future directions or solutions to address this gap. This omission may limit the paper's impact on advancing the field's understanding of integrating these capabilities.
>
>
> We definitely agree with the need to further investigate the results in more domains and examine the broader applicability of the findings. However, we also emphasize that this is the first paper to examine the relationship between these different avenues of reasoning, and thus we emphasized two controlled domains, linear functions and simple translation, in order to obtain more clear-cut and verifiable results. This is akin to the papers examining the expressivity of in-context learning starting with examining linear functions, rather than complex and varied domains, in which results may be unclear for a variety of reasons related to the domain – we did include a real-life task of Kalamang translation, in which we did not see benefits to instruction inference, unlike the first two domains, likely due to a breakdown in hypothesis proposal. Since we have now examined simple domains and have clear results for these simple domains, this makes it easier for future work to build on these results and examine more varied domains while keeping in mind the pieces of the reasoning loop as individual components.
>
> One potential idea is self-consistency training, which could be a form of instruction tuning. For instance, we could use few-shot examples with underlying rules which can be expressed through natural language or code, and have the LM generate both an answer and explanatory rule separately. We can then have a combined loss, a generative loss which is the usual cross-entropy on the answer tokens (with multiple x_is tested for a single rule), and a consistency loss which penalizes the loss incurred by applying the generated rule to the answers generated by the model. We can certainly add more discussion of potential research directions and applications to the conclusion as part of future work.

---

> > ### Comment · Reviewer_z3T2 · 2024-06-04
> >
> > I agree with Reviewer zPDQ that the contributions in the paper seem overclaimed. A title such as "Investigating Instruction Following, Few-Shot Learning, and Instruction Inference..." would be more appropriate.

---

### Official Review · Reviewer_WMHL · 2024-05-12

**Rating:** 7
**Confidence:** 3
**Ethics Flag:** 1

**Summary:**

The paper investigates how modern large language models (LMs) utilize deductive, inductive, and abductive reasoning to learn tasks. The study explores three learning methodologies: instruction following, where tasks are explicitly described; few-shot prompting, where tasks are implied through examples; and instruction inference, where models generate task descriptions from examples before making predictions. Analyzing four LMs from the GPT and Llama families across tasks involving arithmetic functions and machine translation, they find an interesting dissociation in reasoning capabilities. LMs can effectively learn from few-shot examples without being able to articulate the underlying rules, and conversely, they can generate useful task descriptions yet fail to learn from explicit human-generated instructions. The findings suggest that while instruction inference can enhance performance in simpler synthetic setups, the integration of various reasoning types remains challenging in more complex tasks.

**Reasons To Accept:**

The paper is well-organized, and the results are presented clearly.

This paper provides a meaningful first step to understanding the three learning methodologies of LLM.

**Reasons To Reject:**

The paper primarily focuses on specific simple tasks, linear function The scope of tasks examined in the study—linear function learning, artificial language translation, and Kalamang translation—is somewhat narrow, limiting the broader applicability of the findings. Future studies could benefit from incorporating a wider array of tasks to strengthen the arguments presented.

There is room for deeper exploration regarding the limitations of the current methodologies. Discussing potential new domains for future research and outlining prospective directions would provide valuable insights.

---

> ### Author Rebuttal · Authors · 2024-05-30
>
> Thank you for your review, we're glad you appreciate our contributions and writing! See our responses below:
>
> >  The paper primarily focuses on specific simple tasks, linear function The scope of tasks examined in the study—linear function learning, artificial language translation, and Kalamang translation—is somewhat narrow, limiting the broader applicability of the findings. Future studies could benefit from incorporating a wider array of tasks to strengthen the arguments presented.
>
> Thanks for the comment! We agree that there are many potential future applications that are not explored here. As this was a first study exploring the relationship between these types of reasoning in language models, we thought it was important to use clear-cut domains such as the linear functions and colours domains, in which correctness of rules is unambiguous, easing verification. We attempted to show the differences between simple domains and a real-life task by incorporating the Kalamang translation results, in which instruction inference does not provide a large increase in performance, because of the breakdown in grammar rule proposal. We will explore more diverse domains as well as more avenues for improving instruction inference in future work.
>
> > There is room for deeper exploration regarding the limitations of the current methodologies. Discussing potential new domains for future research and outlining prospective directions would provide valuable insights.
>
> This is a good point! We can certainly add more discussion of potential research directions and applications to the conclusion as part of future work. For instance, although the set of tasks we examined is more related to translation, one could imagine that similar methods could be used to help LMs reason about any domain with underlying rules or principles. For instance, LMs following a similar strategy could be useful to aid researchers in proposing scientific hypotheses to test given a body of existing literature.

---

> > ### Comment · Reviewer_WMHL · 2024-06-05
> > **Acknowledgment of Response**
> >
> > After examining your responses as well as the remarks from other reviewers, I agree that the title may be an overstatement. It would be better to consider toning down the title.

---

### Decision · Program_Chairs · 2024-07-10

**Decision:**

Accept

**Comment:**

This paper received two lukewarm and two strong reviews. Based on my reading of the reviews and paper, among the strengths and weaknnesses, I believe the following are relevant:

Strengths:
- The reviewers agreed the problem is well-motivated, and found the paper innovative, interesting, and well-written.

Weaknesses:
- The title is an overstatement (WMHL, z3T2; cf also statement by zPDQ). This is easy to fix: The authors are urged to consider more accurate titles, such as the suggestion by reviewer z3T2.
- While the tasks are diverse (linear functions, artificial languages, Kalamang translation) are quite diverse, the set of tasks is still limited (WMHL).
- No generalizable knowledge (zPDQ) or meaningful takeaways for future work (WMHL, z3T2) are derived from the results. The authors counter that identifying empirical results itself is valuable.

Overall, I concur with the reviewers in an overall positive assessment, and believe the paper meets the high standards of this conference.

[comment from PCs] Please consider the title suggestions of the AC and reviewers.